# Combined Inhibition of Hedgehog and HDAC6: In Vitro and In Vivo Studies Reveal a New Role for Lysosomal Stress in Reducing Glioblastoma Cell Viability

**DOI:** 10.3390/ijms24065771

**Published:** 2023-03-17

**Authors:** Alex Pezzotta, Loredana Brioschi, Sabrina Carbone, Beatrice Mazzoleni, Vittorio Bontempi, Federica Monastra, Laura Mauri, Anna Marozzi, Marina Mione, Anna Pistocchi, Paola Viani

**Affiliations:** 1Department of Medical Biotechnology and Translational Medicine, University of Milan, L.I.T.A., Via Fratelli Cervi 93, Segrate, 20054 Milano, Italy; 2Molecular Mechanisms Unit, Fondazione IRCCS Istituto Nazionale dei Tumori di Milano, Via Giacomo Venezian, 1, 20133 Milano, Italy; 3Department of Cellular, Computational and Integrative Biology (CIBIO), University of Trento, Via Sommarive 9, 38123 Trento, Italy

**Keywords:** glioblastoma, zebrafish, *Hh*, HDAC6, lysosomal stress, autophagy, sphingolipids, combination treatment

## Abstract

Glioblastoma multiforme (GBM) is the most common and malignant brain tumor in adults. The invasiveness and the rapid progression that characterize GBM negatively impact patients’ survival. Temozolomide (TMZ) is currently considered the first-choice chemotherapeutic agent. Unfortunately, over 50% of patients with GBM do not respond to TMZ treatment, and the mutation-prone nature of GBM enables the development of resistance mechanisms. Therefore, efforts have been devoted to the dissection of aberrant pathways involved in GBM insurgence and resistance in order to identify new therapeutic targets. Among them, sphingolipid signaling, Hedgehog (*Hh*) pathway, and the histone deacetylase 6 (HDAC6) activity are frequently dysregulated and may represent key targets to counteract GBM progression. Given the positive correlation between *Hh/HDAC6*/sphingolipid metabolism in GBM, we decided to perform a dual pharmacological inhibition of *Hh* and HDAC6 through cyclopamine and tubastatin A, respectively, in a human GMB cell line and zebrafish embryos. The combined administration of these compounds elicited a more significant reduction of GMB cell viability than did single treatments in vitro and in cells orthotopically transplanted in the zebrafish hindbrain ventricle. We demonstrated, for the first time, that the inhibition of these pathways induces lysosomal stress which results in an impaired fusion of lysosomes with autophagosomes and a block of sphingolipid degradation in GBM cell lines. This condition, which we also recapitulated in zebrafish embryos, suggests an impairment of lysosome-dependent processes involving autophagy and sphingolipid homeostasis and might be instrumental in the reduction of GBM progression.

## 1. Introduction

Glioblastoma multiforme (GBM) is the most common form of brain tumor and accounts for 50% of all glioma cases. This malignant tumor arises from astrocytes, the glial cells which serve supporting roles within the nervous system [1]. GBM is a highly invasive tumor that penetrates into the surrounding brain parenchyma and is usually confined to the central nervous system. To date, GBM patients have a very poor prognosis, which is one of the worst in modern oncology [2,3]. The current treatment for GBM is surgical resection of the tumor area followed by a combination of radio- and chemotherapy. Among the chemotherapeutic agents used, temozolomide (TMZ), a brain-penetrating alkylating agent that methylates purines in DNA and induces apoptosis [4], is the standard choice for GBM. Despite TMZ treatment, GBM recurs and becomes resistant to further chemotherapies, and new therapeutic strategies urgently need to be found [5]. Notably, the current therapies for GBM induce multiple effects on the sphingolipid pathway. Radio- and chemotherapy can enhance proapoptotic ceramide levels, and alteration in the ceramide metabolism has been described as a feature of several cancers that develop resistance to chemotherapeutic agents [6]. It is known that ceramide levels in GBM are inversely associated with tumor progression and poor prognosis. Moreover, human gliomas have been shown to have increased metabolic flux from ceramide to the prosurvival sphingosine 1-phosphate (S1P), suggesting that these modifications in ceramide metabolism may underlie the tumor’s malignant features [7,8]. Apart from metabolic changes, dysregulation of different pathways such as the activation of the Hedgehog (*Hh)* signaling, contributes to GBM progression [9,10]. The higher expression of the Sonic Hedgehog (SHH) ligand in GBM cells, such as U87-MG and primary tumor samples, correlates with the higher expression of downstream genes important for GBM survival [11]. For this reason, several studies investigated the effect of *Hh* inhibition with Smoothened (SMO) antagonists, such as cyclopamine, to counteract GBM progression [12,13]. In addition to *Hh* hyperactivation, the overexpression of Histone deacetylase 6 (HDAC6) in GBM tumor promotes cell proliferation and TMZ resistance [14,15,16], indicating that HDAC6 targeting could represent a possible therapeutic strategy to counteract GBM progression. Indeed, the selective inhibition of HDAC6 through tubastatin A reverted the malignant phenotype of GBM cells in an in vitro study, increasing TMZ-induced apoptosis [17]. However, conflicting evidence exists regarding the role of autophagy in GBM. It is largely recognized that this process is involved in the initiation, development, and response to treatments of this tumor [18]. Both HDAC6 and *Hh* have a prominent role in the regulation of autophagy at multiple levels [19,20]; therefore, the effects of their combined inhibition on this process could be an intriguing feature to examine. Interestingly, we recently demonstrated that there is crosstalk between *Hh* and HDAC6 and that their inhibition effectively reduces the expansion of hematopoietic stem progenitor cells presented in zebrafish models of myeloid leukemia [21]. Zebrafish, (*Danio rerio*), is a well-established animal model used to study human diseases, including cancer. Indeed, there are many aspects that make the zebrafish useful for this purpose. Among these include the genetic manipulation of several genes involved in tumor development, which allows for the generation of tumor-animal models, including of GBM fish. Moreover, zebrafish embryos are suitable for the xenotransplantation of mammalian cells [22,23]. Indeed, cells stained with chemical dyes, such as CM-Dil, and the transparency of the zebrafish embryos allow for the in vivo dynamics of tumor development to be followed [24]. The xenotransplantation of human labeled GBM cells into the brain, in particular the hindbrain ventricle, allows for the orthotopic implantation of these cells and permits the evaluation of the GBM tumor invasion and the response to various drug treatments [25,26].

In this work, we investigated the effects of combined inhibition of HDAC6 and the *Hh* signaling pathway, through tubastatin A (TubA/T) and cyclopamine (cyclo/C), respectively, in U87-MG human cells and zebrafish embryos. We also tested the effects of drug treatments on the growth of cells orthotopically transplanted into the hindbrain of zebrafish embryos. We observed a significant reduction in cell viability in both systems after combination treatment. Mechanistically, this reduction in cell growth was due to a dysfunction in the autophagic process, lysosome engulfment, and sphingolipid metabolism impairment. These autophagy-related defects are specifically caused by *Hh* and HDAC6 hyperactivation, as we detected an increase in the autophagic process both in a zebrafish model for *Hh*/*hdac6* overexpression and in the zebrafish GBM adult brain tissues. We propose the combined inhibition of *Hh* and HDAC6 as a powerful treatment for GBM and our zebrafish models as high-throughput platforms to further identify new effective compounds.

## 2. Results

### 2.1. Combined Inhibition of HDAC6 and the Hh Pathway Synergistically Suppressed U87-MG Cell Viability In Vitro and In Vivo

Data from the literature reported the overexpression of HDAC6 [14,15,27] and the hyperactivation of the *Hh* signaling in GBM [9,11]. By means of the GEPIA2 webtool, we found a positive correlation between *Hh* and *HDAC6* expression in the GBM datasets, suggesting a possible crosstalk between them (Appendix A). Therefore, we tested the effect of HDAC6 and *Hh* inhibition, alone or in combination, on U87-MG glioma cell viability. To this purpose, we administered tubastatin A (TubA/T) as a specific HDAC6 inhibitor and cyclopamine (cyclo/C) as a specific *Hh* signaling inhibitor following the setup of experimental conditions (Appendix A). We then treated cells with 8 μM of TubA and 10 μM of cyclo, alone or in combination, for 48 h. While cyclo administration did not significantly reduce cell viability, TubA treatment elicited a 31% reduction compared to the control. In the combination treatment, cyclo significantly potentiated the TubA-mediated effect, causing a 52% decrease of cell survival (Figure 1A). We also demonstrated that the effect of cyclo and TubA treatments on U87-MG cells persisted up to 1 day after the removal of the two inhibitors (Appendix A). To corroborate this effect in an in vivo microenvironment, we injected labeled U87-MG cells, both pretreated with the two drugs and untreated, into the hindbrain ventricle of 2 dpf zebrafish embryos (Figure 1B–F). The growth capability of U87-MG cells was evaluated by normalizing the total area occupied by the labeled-xenografted cells observed one day after injection (time 1, t1) to the area measured immediately after the implant (time 0, t0; Appendix A). The growth of xenografted U87-MG cells treated with cyclo alone was similar to that of control cells, while TubA treatment elicited a significant reduction of the area occupied by tumor cells (Figure 1B–D,F). However, U87-MG cell growth rate reduction was significantly enhanced when cells were pretreated with cyclo and TubA (Figure 1E,F), supporting the results obtained in the in vitro assays. Since MAPK/ERK and PI3K/Akt are important cellular signaling cascades that control cell growth and survival, we then examined whether the inhibition of HDAC6 and the *Hh* pathway affects ERK and Akt activation (Figure 1G; Appendix A). As shown in Figure 1G, the results demonstrated that ERK phosphorylation was downregulated by TubA both as a single treatment and in combination with cyclo, with a 53% and a 62% decrease compared to the control, respectively. No significant changes were observed with cyclo alone. Similarly, pAKT levels were reduced by TubA alone and in combination with cyclo, with a 36% and a 52% decrease compared to the control, respectively, while cyclo alone did not elicit a significant decrease. These results indicated that the inhibition of HDAC6, but not that of *Hh*, contributed to the downregulation of these pathways.

### 2.2. Combined Inhibition of HDAC6 and the Hh Pathway Impaired Autophagy in U87-MG Cells and Zebrafish Embryos

Autophagy is considered a promising target to counteract GBM progression and, interestingly, is regulated by both HDAC6 and *Hh.* Therefore, we evaluated the effect of TubA and cyclo on this pathway by assessing the expression of the autophagic marker LC3-II, the phosphatidylethanolamine-conjugated form of the cytosolic LC3-I, that indicates the formation of the autophagosome [28]. Cells treated with TubA did not show differences in LC3-II levels compared to the control, while cyclo administration elicited an increase in LC3-II expression of about 5-fold, indicating that cyclo promotes an increased amount of autophagosomes (Figure 2A,B black bars, Appendix A). Interestingly, the combined inhibition of *Hh* and HDAC6 resulted in a 10-fold increase in LC3-II protein production (Figure 2A,B black bars). We further assessed at which step of the autophagic process TubA and cyclo exerted their effects by taking advantage of different autophagy inhibitors: 3-methyladenine (3-MA), which inhibits PI3K-III and is essential for autophagosome membrane nucleation from intracellular membranes; and bafilomycin A1 (BafA1), which blocks later stages of the autophagic process (i.e., autophagic vacuole maturation and degradation). When cells were treated with cyclo alone or in combination with TubA in the presence of 3-MA (Figure 2A grey bars), LC3-II accumulation was completely abolished, indicating that treatment with cyclo alone or in combination acts after the nucleation of the autophagosome. On the contrary, BafA1 promoted a strong accumulation of LC3-II in control and TubA- or cyclo-treated cells, whereas it did not induce any further LC3-II accumulation in the combined treatment, suggesting a full impairment of the later stages of the autophagic flux in the presence of both inhibitors (Figure 2B grey bars). We obtained similar results evaluating the expression of the autophagic substrate p62/SQSTM1 [29]: if autophagy is defective, p62 is not degraded and therefore accumulates. Treatments with cyclo alone and in combination with TubA induced an increase in p62/SQSTM1 expression levels of about 1-fold compared to the control, confirming the block in the autophagic flux exerted by cyclo and combination treatments (Figure 2C).

To dissect the effect of the simultaneous *Hh*/HDAC6 inhibition on the autophagic process in vivo, we used the zebrafish Tg(*CMV:eGFPmap1Lc3b*) line, which allows for the visualization of the Lc3b autophagosomes [30]. Before investigating the effects of TubA and cyclo administration on the autophagic process, we evaluated the response of the Tg(*CMV:eGFPmap1Lc3b*) line to well-established modulators of this process. According to data from the literature, the administration of rapamycin (rap), an activator of the autophagic flux, significantly reduced the number of Lc3b autophagosomes, suggesting a faster rate of the autophagic flux (Appendix A). The treatment with BafA1 led to an increased number of autophagosomes, indicating an autophagic block (Appendix A), thus mimicking the condition obtained in U87-MG cells. Moreover, we performed a dose-response assay to select the working doses for TubA and cyclo. We verified that the administration of 5 μM of cyclo and 25 μM of TubA, alone or in combination, dissolved directly into zebrafish embryos’ growth medium from 1 dpf to 2 dpf did not alter the whole-embryo development. Therefore, we selected these doses for further experiments (Appendix A). Immunofluorescence analyses showed that the number of Lc3b autophagosomes was not significantly altered in embryos grown in the medium containing cyclo or TubA compared to controls treated with the vehicle DMSO (Figure 2D–G,I). On the contrary, the coadministration of these compounds elicited a significant increase of Lc3b autophagosomes in zebrafish embryos (Figure 2H,I). Moreover, in embryos treated with cyclo or TubA alone, the colocalization of Lc3b/p62 was not different from that observed in the DMSO-treated control embryos (Figure 2J–L,N). In contrast, embryos treated with both compounds showed a significant increase of the p62 signal into Lc3b autophagosomes (Figure 2M,N), a situation that recapitulates the blockade of the autophagic process observed in the U87-MG cell lines treated with TubA and cyclo.

### 2.3. Combined Inhibition of the HDAC6 and Hh Pathway Induced Lysosome Alteration in U87-MG Cells and Zebrafish Embryos

To evaluate whether the observed changes in autophagic flux are due to alterations of lysosomes, we assessed the number of acidic compartments in control and treated U87-MG cells using acridine orange (AO) staining (Figure 3A). Even though all treatments promoted AO staining, the higher increase was observed with the combined inhibition of HDAC6 and *Hh*. We also induced an increase in the number of lysosomes in U87-MG cells following single or combination treatments by using a specific dye for lysosomes: Lysotracker Red DND-99 (Figure 3B). Furthermore, the expression of the lysosomal marker LAMP1, analyzed using Western blot techniques, was increased to 30% following combination treatments compared to the control (Figure 3C). Since the administration of cyclo and TubA increased the acidic compartment in vitro, we further investigated the lysosomal compartment in vivo using LysoTracker Red DND-99 (Figure 3D). Lysosome staining in zebrafish embryos with TubA alone was similar to that found in Dimethyl sulfoxide(DMSO) control embryos (Figure 3E–G,I). Conversely, the fluorescence intensity of the LysoTracker signal was increased following administration of cyclo and mostly in the combination treatment (Figure 3H,I). Altogether, these results indicated that the combination treatment altered the lysosomal compartment in both the in vitro and in vivo models.

### 2.4. Combined Inhibition of HDAC6 and Hh Pathway Impaired the Lysosomal Catabolism of Complex Sphingolipids

In silico analyses on GBM datasets demonstrated a positive correlation between the expression levels of the *Hh* signaling, HDAC6, and the main enzymes involved in the sphingolipid metabolism (Appendix A). To evaluate the potential effect of TubA and cyclo on sphingolipids levels, we performed a pulse-chase experiment with [^3^H]Sphingosine ([^3^H]Sph) at the metabolic equilibrium. [^3^H] Sph can be used to label all sphingolipids, as it is rapidly internalized in the cells and efficiently converted into ceramide (Cer), which in turn is used in the biosynthetic pathway to generate sphingomyelin (SM) and glucosylceramide and then converted to more complex glycosphingolipids (GSL). After a 2 h pulse with [^3^H]Sph and a 24 h chase to reach the steady-state labeling, cells were treated with TubA and cyclo, alone or in combination for 48 h. The combined treatment significantly modified the radioactivity distribution among the different sphingolipid metabolites, causing an accumulation of [^3^H]Cer and all complex [^3^H]sphingolipids (Figure 4A). In particular, the radioactivity associated with Cer, SM, and GSL was increased by about 240%, 130%, and 45%, respectively, in the cotreated cells compared to the control. TubA alone also caused an increment in [^3^H]SM and [^3^H]GSL levels without affecting [^3^H]Cer, while no significant changes were detectable after cyclo treatment. Interestingly, even no significant differences were observed in sphingosine 1-phosphate (S1P) levels in any of the tested conditions, the S1P/Cer ratio was significantly reduced following all treatments, with a maximum of a 3-fold decrease in the presence of combination treatment, indicating a shift of S1P-Cer intracellular balance in favor of Cer. These results suggested a reduced lysosomal activity that strongly affected sphingolipid metabolism and levels.

For this reason, SM hydrolysis into Cer was assessed in experimental conditions in which exogenously added radioactive SM was internalized by endocytosis and degraded in lysosomes after endosome–lysosome fusion. Cells were first treated with TubA and cyclo alone or in combination and then pulsed with radioactive SM. As shown in Figure 4B, at the end of the 2 h pulse, the incorporated radioactivity was reduced in TubA and cotreated cells, indicating an inhibitory effect of TubA alone or in combination with cyclo on endosome formation and internalization. Moreover, the analysis of radioactivity associated with Cer and SM indicated a significant reduction of [^3^H]Cer formation in combination treatments compared to the control cells and a more limited effect in TubA-treated ones. In contrast, no significant differences in the amount of radioactive SM were observed in any of the treatments. To study the effect of combined inhibition of *Hh* and HDAC6 on the internalization and intracellular trafficking of SM, we used BODIPY-SM, a fluorescent derivative of SM under experimental conditions that first allowed SM insertion into the plasma membrane and then its internalization via endocytosis. After 30 min of incubation at 4 °C, BODIPY-SM associated almost exclusively with the plasma membrane in both control and treated cells (Figure 4C). The distribution of fluorescence in the membrane was fairly homogeneous in control cells, whereas in the combination treatment, the fluorescence was present as distinct and more intense dots throughout the membrane. At 37 °C in control cells, most of the fluorescence gradually accumulated in a broad perinuclear region (left panel Figure 4C), representative of the lysosomal compartment; at 60 min of incubation, a partial recovery of fluorescence was observed at the plasma membrane level, indicating a recycling of fluorescent fatty acid generated in the lysosomes for membrane lipid resynthesis. Conversely, in combination treatment at all times considered, fluorescence spread as aggregates throughout the cytoplasm (right panel Figure 4C), thus indicating an impaired metabolic turnover of membrane SM after endocytosis, probably due to an altered trafficking of endosomes to lysosomes. All these data indicated a reduced lysosomal activity in the presence of combination treatment and, to a lesser extent, with TubA.

### 2.5. Tubastatin A and Cyclopamine Prevented Autophagosome and Endosome Fusion with Lysosomes

The accumulation of autophagosomes and lysosomes in both U87-MG cells and zebrafish embryos and the altered subcellular distribution of fluorescent SM induced by combination treatments in U87-MG cells prompted us to explore whether a combined inhibition of *Hh* and HDAC6 could alter the fusion of lysosomes with autophagosomes or membrane-generated endosomes. In control cells, LC3-associated fluorescence assessed with immunofluorescence staining was almost homogenously diffused, whereas the accumulation of LC3 puncta, indicating the presence of autophagic vacuoles in the cytoplasm, was observed following the combination (Figure 5A) treatments. Staining of LAMP1 indicated a prevailing perinuclear localization of these organelles in control cells and an increase in the number and diffusion of lysosomes in the cytoplasm of treated cells (Figure 5A). The fusion of autophagosomes (green) with lysosomes (red), should result in the colocalization of the two markers, subsequently giving rise to orange-yellow staining in merged images. The merging of LC3 and LAMP1 staining following combination treatments showed the separation among LC3 (green) and LAMP1 (red) staining, indicating a lack of lysosome–autophagosome fusion. These results collectively indicate that following combination treatments, autophagosomes were unable to fuse with lysosomes, thus leading to a block in the autophagic flux and to the accumulation of autophagic vacuoles.

To investigate the possible effect of combination treatments in targeting SM-containing endosomes to lysosomes in U87-MG cells, we analyzed fluorescence distribution in the control and treated cells labeled with BODIPY-SM and Lysotracker Red (Figure 5B). In control cells after 30 min incubation, the fluorescence of BODIPY-SM and that of LysoTracker Red almost completely colocalized in the perinuclear region indicating that endocytosed fluorescent SM was correctly addressed to lysosomes. In treated cells, the distribution of BODIPY-SM and LysoTracker Red was diffused throughout the cytoplasm without colocalization as indicated by the presence in the merged images of distinct green and red dots representative of endosomes and lysosomes, respectively.

### 2.6. In Vivo Zebrafish Models of GBM Recapitulated the Alteration in the Autophagic Process Shown in U87-MG Cells

To determine if the effect of the simultaneous *Hh*/HDAC6 inhibition on the autophagic process are specifically due to these pathways, we used our previously generated zebrafish model in conjunction with hyperactivation of the *Hh* signaling. Indeed, we demonstrated the existence of a crosstalk between these two signaling pathways, as the overexpression of the *Hh* ligand *shh* also elicited the upregulation of *hdac6* [21]. Using this in vivo model, we verified that the number of Lc3b autophagosomes was significantly decreased in the *shh*-mRNA-injected embryos from the Tg(*CMV:eGFPmap1Lc3b*) line as compared to the control embryos treated with the DMSO (Figure 6A–C). These results confirmed that the hyperactivation of the *Hh* signaling, and in turn, the increased expression of *HDAC6* present in GBM tumor setting, were sufficient to activate the autophagic process, and, therefore, the combined inhibition of these pathways might be efficient in blocking it, leading to an engulfment of the cell. To further confirm the role of *Hh* pathway activation and autophagy in vivo and in a GBM condition, we evaluated the expression levels of the *Hh* downstream effector *gli1* and those of some genes necessary for the activation of the autophagic process in a zebrafish model of GBM generated and described by Mayrhofer and collaborators [31]. Indeed, in this GBM zebrafish model, the activation of the EGFR/RAS/ERK/AKT pathway through the zic4 enhancer induces brain tumor development with a mesenchymal GBM signature. The analysis of the RNAseq data (GSE74754) confirmed that the hyperactivation of the *Hh* signaling (i.e., *gli1* expression) (Figure 6D) was accompanied by the increased expression of *atg4a*, *atg4c*, and *dram1* (Figure 6E–G) in the brain of zicras transgenic fish compared to wt control. These data were in line with the autophagy gene expression profiles found in the GBM datasets that exhibit increased transcription of ATG genes and DRAM [32] as well as the increase of Lc3b autophagosomes following *Hh* hyperactivation, supporting the role of this signaling pathway as an activator of the autophagic process in a GBM-like condition.

## 3. Discussion

GBM is characterized by aggressiveness, recurrence, and poor prognosis. The impossibility of extensive tumor removal and the failure of most of the pharmacological treatments to exert antitumor effects due to their difficulty in crossing the blood–brain barrier, contribute to the lack of effective treatments. Therefore, new therapeutic approaches are urgently needed to prolong patients’ survival. Multidisciplinary studies combining different model systems could improve efforts in this direction. The use of animal models represents a powerful strategy to mimic tumor growth in an in vivo microenvironment. For instance, it is possible to generate knock-down, knock-out, or overexpressed zebrafish models to the mimic molecular pathways found altered in individuals with GBM [33].

Among them, the overexpression of HDAC6 and the abnormal activation of the *Hh* pathway lead to GBM resistance to temozolomide (TMZ) [10,15]. Indeed, the *Hh* pathway regulates *O-6-methylguanine-DNA methyltransferase* (*MGMT*) expression and influences the response to TMZ treatment [34]. Similarly, overexpression of HDAC6 leads to resistance to TMZ by reducing the activity of the mismatch repair (MMR) pathway [35]. Several studies investigated the potential of HDAC6 and *Hh* pathway single inhibition. For example, selective inhibition of HDAC6 with TubA reverses the malignant phenotype of GBM cells by increasing TMZ-induced apoptosis [17] while the blockade of *Hh* signal prevents glioma stem cell (GSCs) survival [10]. In the literature, a mechanism by which HDAC6 inhibition leads to the modulation of the *Hh* signaling has been described. Indeed, HDAC6 blockade downregulates *Hh* target GLI1 and PTCH1/2 expression and activity and increases apoptosis of GSCs [36]. Previous works have demonstrated that the combined inhibition of HDAC6 and *Hh* pathway using TubA and cyclo treatment effectively reduces glioma cells’ clonogenicity and migration capacity [17,37]. However, the mechanism of the combined inhibition of TubA and cyclo on GBM cells has never been investigated.

In this work, we attempted to fill this gap in the literature by providing data supporting the potential of *Hh* and HDAC6 combined inhibition as a novel therapeutic strategy for treating patients with GBM. By testing the combined inhibition of *Hh* and HDAC6, we showed a more significant reduction of U87-MG cell viability than that yielded with single *Hh* or HDAC6 inhibition. Interestingly, we showed that the effects of the combined treatment on cell viability were also maintained in a microenvironment that mimicked the complexity of the human one, such as the hindbrain ventricle of zebrafish embryos. The zebrafish provides a powerful platform for xenotransplantation of human GBM cells, as the adaptive immune system develops after the first month of life, ensuring remarkable engraftment of human cancer cells during this developmental window [38]. Moreover, the xenotransplantation of patient-derived cancer cells enables zebrafish to be employed in the field of personalized medicine. Indeed, treating these xenografted embryos will lead to the establishment of patient-specific therapy [39].

The first pathways understood to constitute the mechanism underlying the observed effect of the TubA and cyclo on U87-MG cell viability was MAPK/ERK, essential for cell proliferation and transformation [40], and PI3K/Akt, which is often hyperactivated in GBM and responsible for cell survival and drug resistance [41]. The inhibition of HDAC6 but not that of the *Hh* pathway downregulated MAPK and PI3K/Akt pathways in U87-MG cells, thus indicating that they are not involved in the synergism observed following the combination treatment.

Autophagy is a process that plays a significant role in GBM homeostasis [18], and both the HDAC6 and the *Hh* pathways are known to have a prominent function in this process [20]. For instance, HDAC6 regulates this process with different mechanisms that span from the post-translational modification of autophagy-related proteins to the transport of autophagosomes and their fusion with lysosomes [20]. On the contrary, the role of the *Hh* signaling pathway varies according to the cancer type, acting as an inhibitory modulator as well as a promoting factor of the autophagic process [42]. Many studies in GBM have shown that autophagy inhibition enhances the radiosensitivity and chemosensitivity of glioblastoma cells [43]. On the other hand, autophagy induction can play a positive role by inhibiting tumor formation and progression [44]. Therefore, a treatment combining autophagy inducers and inhibitors could be a feasible strategy to improve therapeutic effects.

In this work, using both in vivo and in vitro models, we demonstrated that combined inhibition of *Hh* and HDAC6 alters autophagy. In particular, in U87-MG cells, we demonstrated that inhibition of *Hh* by cyclo alone or in combination promotes an increased autophagosome synthesis and an alteration of autophagosome maturation, as also observed in Hela cells [45]. Nevertheless, HDAC6 inhibition alone does not affect the autophagic pathway but strongly enhances cyclo-dependent autophagosome accumulation. The dual effect of *Hh* and HDAC6 combined inhibition on autophagy was recapitulated in treated Tg(*CMV:eGFP-map1Lc3b*) zebrafish strains, demonstrating the capacity of this model to be a valid tool in studying the autophagic process. In addition, by expression analysis of the *Hh* signaling- and autophagy-related genes in brain tumors of a GBM zebrafish model and the evaluation of the Lc3b signal in the model with *Hh*/Hdac6 dysregulation, we provided data supporting the effects of the combination of *Hh* and HDAC6 inhibition in cancer treatment.

Our results obtained in vitro and in vivo strongly indicate the existence of a block at the level of autophagosomes clearance, possibly dependent on the lysosomal function. We demonstrated that U87-MG cells and zebrafish embryos treated with both TubA and cyclo show a marked increase in the number of acidic compartments, specifically lysosomes, that could be indicative of either an altered lysosomal turnover or increased biogenesis, which is known to be a compensatory mechanism to counteract lysosomal stress [46]. Interestingly, the increased number of lysosomes was also associated with an alteration of sphingolipid metabolism and levels in U87-MG cells. In fact, HDAC6 and *Hh* combined inhibition causes a nonspecific accumulation of ceramide and almost all complex sphingolipids associated with impaired degradation and recycling of the plasma membrane SM. We also observed a reduction of incorporated radioactivity after TubA treatment alone and combined with cyclo, indicating an alteration of the endocytic pathway. This is not unexpected, given that HDAC6 has a prominent role on cortactin’s deacetylation and that this protein’s hyperacetylation can prevent its association with F-actin, thus impairing optimal endocytosis [47,48].

These effects demonstrated an impairment of the lysosomal catabolic function that results in a block of the recycling pathway, the main active pathway responsible for the synthesis of new membrane sphingolipids in highly proliferating cells, and a shift of the S1P/Cer rheostat towards the prodeath ceramide. Samples from GBM patients have demonstrated an increased content of S1P with a concurrent decrease in ceramide compared to that from normal brains. Notably, the current therapies for GBM work in part by altering sphingolipid metabolism to enhance ceramide levels [49,50].

Overall, these results demonstrated that HDAC6 and *Hh* combined inhibition promotes an accumulation of both autophagic and metabolic substrates due to the lack of a correct degradation in the lysosomes. In U87-MG cells cotreated with Tuba and cyclo, autophagosomes and endosomes fail to reach and fuse with lysosomes, thus preventing their degradation and turnover. A major role in the impaired transport to lysosomes could be ascribed to HDAC6 inhibition due to its role in the transport of lysosomes along microtubules to promote autophagic substrate degradation [20]. We propose a molecular mechanism, shown in Figure 7, that describes how the *Hh* and HDAC6 inhibition might alter the trafficking of endosomes and autophagosomes to lysosomes. These data highlight the role of the double inhibition of HDAC6 and *Hh* in regulating endosomal and autophagic-lysosomal system functionality, thus affecting multiple cellular processes. 

Further studies need to be carried out to determine if the impairment in the autophagic flux could be linked to dysfunction in the formation of the autophagosome. It could be interesting to evaluate the upstream autophagic markers, such as the Beclin 1-PI3KC3 complex, a lipid–kinase complex involved in autophagosome nucleation.

Therefore, to better characterize the crosstalk between Hh, HDAC6, and autophagy, it would be of interest to replicate the experiments in a well-established zebrafish GBM model [31]. The possibility to validate the effect of the combined treatment at the level of the autophagy flux in an in vivo model of GBM tumor may potentiate the role of zebrafish as a screening platform and as a tool to discover the mechanisms at the basis of several therapeutic strategies.

Our insights related to the molecular mechanism connecting *Hh,* HDAC6, and GBM progression could be translated to other tumors in addition to GBM. Indeed, several cancer types, such as acute myeloid leukemia and colon cancers, overexpress these pathways. Moreover, using zebrafish as tumor models or performing orthotopical transplantation of patients’ derived cells will allow for the quick and high-throughput screening of compounds facilitating preclinical analyses and rapidly enabling the identification of new therapeutic targets.

## 4. Materials and Methods

### 4.1. Cell Culture

U87-MG human glioblastoma multiforme cell line, obtained from the Istituto Zooprofilattico Sperimentale della Lombardia e dell’Emilia (Brescia, Italy) was cultured at 37 °C in a 5% CO_2_ humidified atmosphere in Dulbecco’s modified Eagle’s medium, (DMEM) containing, 2 mM of L-glutamine, 100 units/mL of penicillin, 100μg/mL of streptomycin, 0.25 μg/mL of amphotericin B (all purchased from Merck KGaA, Darmstadt, Germany), and 10% (*v*/*v*) fetal bovine serum (FBS; Euroclone S.p.A., Pero, Milan, Italy).

### 4.2. Animals

Zebrafish embryos were raised and maintained under standard conditions according to the national guidelines (Italian decree 4 March 2014, n. 26). Embryos from AB and Tg(*CMV:eGFP-map1Lc3b*) [30] strains were collected by natural spawning, staged according to the reference guidelines, and raised at 28 °C in E3 medium fish water (instant ocean, 0.1% methylene blue in petri dishes). At 24 h postfertilization (hpf), 0.003% 1-phenyl-2-thiourea (PTU; Merck KGaA, Darmstadt, Germany) was added to prevent pigmentation. Before manipulations, embryos were dechorionated and anesthetized with 0.016% tricaine (ethyl 3-aminobenzoate methanesulfonate salt; Merck KGaA, Darmstadt, Germany).

### 4.3. mRNA Microinjection

Zebrafish *shh*-mRNA was in vitro transcribed as previously described [21]. In this process, 100 pg/embryo of the *shh*-mRNA was injected into one-cell-stage of the Tg(*CMV:eGFP-map1Lc3b*) embryos together with the phenol red vital dye as a tracer of microinjection efficacy. As a control, embryos were injected with the same amount of *rfp*-mRNA. 

### 4.4. Pharmacological Treatments

For cell treatments, tubastatin A (TubA; Merck KGaA, Darmstadt, Germany) and Bafilomycin-A1 (BafA1; MedChemExpress, Princeton, NJ, USA) were dissolved in dimethyl-sulfoxide (DMSO). Cyclopamine (cyclo; Merck KGaA, Darmstadt, Germany) was resuspended in ethanol. 3-methyladenine (3-MA; Cayman Chemical Co, Ann Arbor MI, USA) was dissolved in pure water. In all of the experiments. the vehicle’s final concentration never exceeded 0.1% (*v*/*v*). U87-MG cells were seeded at the density of 3300 cells/well in 96-well plates and treated for 48 h in a single or combination setting and in the presence or absence of the following selected subtoxic doses of TubA (8 μM), cyclo (10 μM), BafA1 (30 μM), and 3-MA (2 mM). In zebrafish, TubA, cyclo, rapamycin (Rap; MedChemExpress, Princeton NJ, USA), and BafA1 were dissolved in DMSO. Treatments were performed in a 24-well plate, with a maximum of 15 embryos/well from the stage at 1 day postfertilization (dpf) to 2 dpf. Tg(*CMV:eGFP-map1Lc3b*) embryos were treated with 25 μM of TubA and 5 μM of cyclo, alone or in combination setting, in 1 mL of final volume of E3 with PTU 1X. The dose–response assay for TubA and cyclo and the modulation of autophagy with 1 μM of rapamicyn (Rap) or 20 nM of BafA1 are shown in the Appendix A.

### 4.5. Cell Viability Assays

U87-MG cells were seeded at a density of 3.3 × 10^4^ cells/well in 96-well plates. After treatments, cell viability was assessed with MTT (3-(4,5-Dimethylthiazol-2-yl)-2,5-diphenyltetrazolium bromide), and 50 µL/well of 0.0 5% MTT (Merck KGaA, Darmstadt, Germany) in DMEM supplemented with 10% FBS was added to each well. Following a 4-hour incubation at 37 °C and 5% CO_2_, cells were lysed with 50 µL/well of a solution containing 80 % isopropyl alcohol, 10% Triton X100, and 10% HCl 0.1N. Absorbance was measured at 570 nm with the Multilabel Counter Wallac Victor 2 (PerkinElmer, Waltham, MA, USA).

### 4.6. Xenograft of U87-MG Treated Cells

Treated U87-MG cells were labeled with the CM-DiI red fluorescent dye (Thermo Fisher Scientific Inc. Waltham, MA, USA) [51]. After being labeled, cells were washed and incubated in DMEM supplemented with 10% FBS for 2 h at 37 °C and 5% CO_2_. Cells were trypsinized and centrifuged for 10 min at 200× *g*, with the supernatant subsequently being discarded. The cell pellet was washed and resuspended in phosphate-buffered saline (PBS) at a pH of 7.4, (2.5 × 10^5^ cells/μL) for injection into the embryos. Nearly 100 GBM cells were microinjected into the hindbrain ventricle of 2 dpf anesthetized embryos of the AB wild-type strain. After injection, embryos were transferred to fresh E3 fish water with PTU 1X and incubated at 32 °C for the rest of the experiment. Images were acquired under fluorescence microscopy (Leica MZ FLIII) on the day of the injection (hereafter referred to as time 0, t0) and one day postinjection (hereafter referred to as time 1, t1). In each experimental condition, the area of a single GBM xenografted embryo at 1 dpi was normalized to the mean area of xenografted embryos belonging to the same experimental group at t0.

### 4.7. Protein Quantification and Immunoblotting

Cells were plated at 10.3 × 10^4^ cells/cm^2^ in DMEM supplemented with 10% FBS. After treatments, cells were washed 2 times with PBS at a pH of 7.4 at 4 °C and then lysed in 20 mM of Tris-HCl pH 7.4, 150 mM of NaCl, 1% Nonidet P-40 (NP-40), 1 mM of NaF, 1 mM of Na_3_VO_4_, 10 mM of sodium pyrophosphate, 1 mM of phenylmethylsulfonyl fluoride (PMSF), and 2 μg/mL each of aprotinin, leupeptin and pepstatin (all purchased from Merck KGaA Darmstadt, Germany). The total protein amount was measured according to the Bradford assay method [52] with Coomassie Brilliant Blue G-250 Dye reagent (Thermo Fisher Scientific Inc. Waltham, MA, USA) and bovine serum albumin as standard. Absorbance was measured at 595 nm. Usually, 25 μg of protein was separated by SDS-PAGE on either 10% (pAKT, AKT, pERK, ERK, LAMP1, GAPDH) or 14% (LC3B and p62) polyacrylamide resolving gels and transferred to nitrocellulose membranes (Bio-Rad Laboratories, Inc. Hercules, CA, USA). Membranes were blocked for 1 h at room temperature with 25 mM of Tris-HCl pH 7.4, 150 mM NaCl (TBS)+ 0.1% Tween-20 (TBS-T) containing 5% (*w*/*v*) nonfat dry milk. Membranes were probed overnight at 4 °C with primary antibodies (Appendix A diluted in blocking solution. After TBS-T washes, membranes were incubated with secondary antibodies (Appendix A) for 1 h at room temperature in blocking solution. Membranes were then washed as above and protein bands were visualized through the UVITEC imaging system using Westar Nova 2.0 Pico Substrate (Cyanagen, Bologna, Italy) or LiteAblot TURBO Extra sensitive chemiluminescent substrate (Euroclone S.p.A., Pero, Milan, Italy).

### 4.8. Metabolic Labeling of U87-MG Sphingolipids with [1-^3^H] Sphingosine

U87-MG cells plated at 10.3 × 10^4^ cell/cm^2^ were maintained 24 h in DMEM with 10% FBS and then loaded with [^3^H]Sph at the final concentration of 0.5 µCi/mL. After a 2 h pulse, the [^3^H]Sph containing medium from each plate was collected, and fresh complete medium was added for the chase. After a 24 h chase, the medium was collected again and replaced with the conditioned medium (obtained from cells cultured in the same conditions but in the absence of [^3^H]Sph) containing 8 μM of TubA, 10 μM of cyclo, and their combination or the vehicle. After 48 h incubation, cells were washed with PBS at 4 °C and harvested, with the total lipids being extracted and processed as previously described [53] Radioactive sphingolipids were separated using HPTLC with different solvent systems, and HPTLC plates (Merck KGaA, Darmstadt, Germany) were then subjected to digital autoradiography with a Beta-Imager 2000 instrument (Biospace, Paris, France). The radioactivity associated with individual lipids was determined using the M3-Vision software provided with the instrument. [^3^H]-sphingolipids were identified by comigration with standards chromatographed in the same plate.

### 4.9. [Sph-^3^H] Sphingomyelin Metabolism

U87-MG cells plated at 10.3 × 10^4^ cell/cm^2^ were maintained for 24 h in DMEM plus 10% FBS. Stock solution of [Sph-^3^H] Sphingomyelin ([Sph-^3^H]SM) in absolute ethanol was prepared and added to fresh medium. The cells were pulsed for 2 h with [Sph-^3^H]SM (1 μCi/mL), with or without 8 μM of TubA, 10 μM of cyclo, and their combination. At the end, cells were washed twice with PBS at a pH of 7.4 at 4 °C and harvested. Total lipids were extracted and processed as described in 4.8.

### 4.10. Lysotracker Assays

U87-MG cells were plated in 35 mm Petri dishes at 10.3 × 10^4^ cell/cm^2^ and treated as described above. After treatments, cells were stained with 50 nM of LysoTracker Red DND-99 (Thermo Fisher Scientific Inc., Waltham, MA, USA) for 15 min at 37 °C and 5% CO_2_. Then, the medium was replaced with fresh DMEM with 10% FBS, and the images were acquired with an inverted fluorescence microscope (IX 50 Olympus). In zebrafish, LysoTracker Red was added to the embryo medium at the final concentration of 10 μM. According to manufacturer’s instructions, embryos were incubated at 28.5 °C for 45 min and washed three times with 1 mL of fresh E3 fish water with PTU 1X. LysoTracker staining was visualized through fluorescence microscopy (Leica MZ FLIII).

### 4.11. Acridine Orange Staining

U87-MG cells, plated in 35 mm Petri dishes at 10.3 × 10^4^ cell/cm^2^, were treated as previously reported and thereafter incubated with 2 µM of Acridine Orange (Merck KGaA, Darmstadt, Germany) for 20 min at 37 °C and 5% CO_2_. After washes in PBS at a pH of 7.4, cells were fixed with 0.5% glutaraldehyde for 10 min at 4 °C and subsequently washed three times with PBS at 4 °C. Finally, coverslips were mounted on glass slides, and images were acquired with a fluorescence microscope.

### 4.12. Uptake and Intracellular Distribution of BODIPY-C5-Sphingomyelin

U87-MG cells plated at 6 × 10^4^ cell/cm^2^ were grown on a glass coverslip and maintained for 24 h in DMEM with 10%FBS. After 48 h of treatments with or without 8 µM of TubA plus 10 µM of cyclo, cells were incubated at 4 °C for 30 min with 5 μM of N-(4,4-Difluoro-5,7-Dimethyl-4-Bora-3a,4a-Diaza-s-Indacene-3-Pentanoyl) Sphingosyl Phosphocholine, (BODIPY-C5-SM; Thermo Fisher Scientific Inc. Waltham, MA, USA) as a complex with fatty acid-free BSA (1:1, mol:mol) in Krebs Ringer buffer at a pH of 7.4. At the end of loading, after two washes with DMEM with 10% FBS and 0.34 mg/mL of fatty acid-free BSA (Merck KGaA Darmstadt, Germany), cells were incubated for different durations at 37 °C. Cells were then washed three times with PBS and fixed with 0.5% glutaraldehyde solution for 10 min at 4 °C. After washes in PBS at a pH of 7.4 at 4 °C, the specimens were immediately observed and analyzed with a fluorescence microscope (Olympus BX-50).

### 4.13. Immunocytochemistry and Immunofluorescence

Treated U87-MG cells were washed with PBS at a pH of 7.4 and fixed with 0.5% glutaraldehyde for 10 min at 4 °C. After permeabilization with 0.1% Triton-X for 3 min at room temperature, cells were blocked with 10% FBS in PBS at pH of 7.4 for 30 min. Cells were incubated with the primary antibodies (Appendix A) for 17 hours in a moist and humid chamber and then for 1 h at room temperature with the secondary antibodies (Appendix A). Glass slides were mounted using Fluoromount G mounting medium (Immunological Sciences, Roma, Italy). The coverslips were then imaged with a fluorescence microscope (Olympus BX-50). Zebrafish PTU-treated embryos of the Tg(*CMV:eGFP-map1Lc3b*) [30] line were fixed overnight in 4% paraformaldehyde (Merck KGaA, Darmstadt, Germany) in PBS at 4 °C. After 2 h in blocking solution at room temperature, embryos were incubated overnight with the primary antibodies and then for 1 h at room temperature with the secondary antibodies listed in Appendix A. Staining was evaluated using confocal analyses (Leica; TCS-SPII).

### 4.14. Image Processing and Statistical Analyses

Images acquired through fluorescence or confocal microscopy were processed through the use of the ImageJ software (National Institutes of Health, Bethesda, MD, USA), with the proper supplied plugin being applied. All results are presented as mean ± standard deviation (SD). The statistical significance of the data was determined by one-way ANOVA with Tukey post hoc correction or unpaired t-test with Welch correction using GraphPad Prism (GraphPad Software, San Diego, California USA), with *p* ≤ 0.05 (*), *p* ≤ 0.01 (**), and *p* ≤ 0.001 (***) being considered statistically significant values.

### 4.15. GBM Data Analysis

Preprocessed data (count files) of wild-type and zicras (tumor) zebrafish brain were retrieved from the GEO database-GSE74754 [33]. Normalization with the TMM method was performed with the edgeR package in Bioconductor (https://bioconductor.org, accessed on 20 January 2023) to obtain the count per million (cpm) values for the genes of interests. Assembly Zv9 Ensembl 75 of the Danio Rerio genome was used for the annotation. Statistical analysis was performed by applying the unpaired t-test with Welch correction.

## Figures and Tables

**Figure 1 ijms-24-05771-f001:**
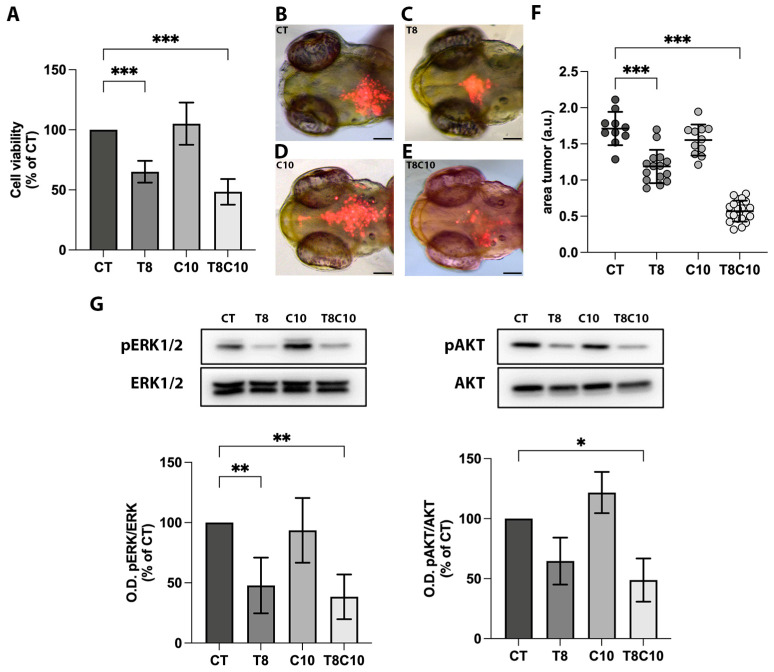
*Hh* and HDAC6 inhibition impaired U87-MG cell viability. (**A**) MTT assay of U87-MG cells treated for 48 h with vehicle (CT), 8 μM of TubA (T8), 10 μM of cyclo (C10) and 8 μM of TubA + 10 μM cyclo (T8C10). (**B**–**E**) Representative images of the head region of 3 dpf zebrafish embryos xenotransplanted with labeled U87-MG cells pretreated with (**B**) CT, (**C**) T8, (**D**) C10, and (**E**) T8C10. (**F**) Quantification of the tumor area at t1 (24 h postinjection, hpi) normalized to the tumor area at t0 (immediately after U87 injection). (**G**) Western blot analyses and quantification of pERK and pAKT protein expression levels following 48 h treatments with or without T8 and C10, alone or in combination. Data are expressed as the percentage of the control (**A**,**G**) or mean ± standard deviation (**F**). CT—vehicle; TubA/T—tubastatin A; cyclo/C—cyclopamine. One-way ANOVA with Tukey post hoc correction. *** *p* < 0.001; ** *p* < 0.01; * *p* < 0.05; nonsignificant data are not shown. Scale bare indicates 100 μm.

**Figure 2 ijms-24-05771-f002:**
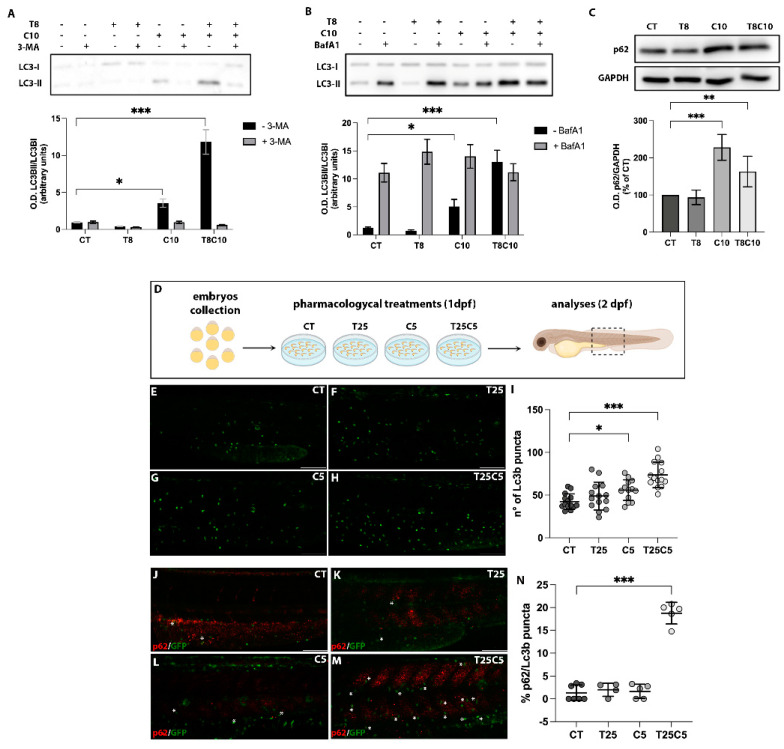
Autophagy was impaired following combination treatments with inhibitors of HDAC6 and *Hh* pathway in U87-MG cells and zebrafish embryos. (**A**,**B**) Western blot analysis and quantification of LC3 protein expression levels after treatments with 8 μM of TubA (T8), 10 μM of cyclo (C10), and 8 μM of TubA + 10 μM of cyclo (T8C10) in the presence (grey)/absence (black) of (**A**) 3-MA or (**B**) BafA1. (**C**) Western blot analysis and quantification of p62/SQSTM1 protein expression following 48 h treatments with or without T8 and C10, alone or in combination (T8C10). Data are expressed as the percentage of the control. (**D**) Schematic representation of the pharmacological treatments. (**E**–**H**) Confocal images of the trunk region of the Tg(*CMV:eGFPmap1Lc3b*) zebrafish embryos at 2 dpf after treatment with (**E**) DMSO, (**F**) 25 μM of TubA (T25), (**G**) 5 μM of cyclo (C5), and (**H**) 25 μM of TubA + 5 μM cyclo (T25C5). (**I**) Count of Lc3b puncta (green puncta) in the trunk region of treated embryos. (**J**–**M**) Confocal images of the trunk region of the Tg(*CMV:eGFPmap1Lc3b*) zebrafish embryos at 2 dpf stained with p62 antibody after treatment with (**J**) DMSO, (**K**) T25, (**L**) C5, and (**M**) T25C5. (**N**) Percentage of Lc3b/p62 puncta in the trunk region of treated embryos. Asterisks indicate the colocalization. Scale bar indicates 100 μm. CT—vehicle; TubA/T—tubastatin A; cyclo/C—cyclopamine. One-way ANOVA with Tukey post hoc test. Data are presented as mean ± standard deviation *** *p* < 0.001; ** *p* < 0.01; * *p* < 0.05.

**Figure 3 ijms-24-05771-f003:**
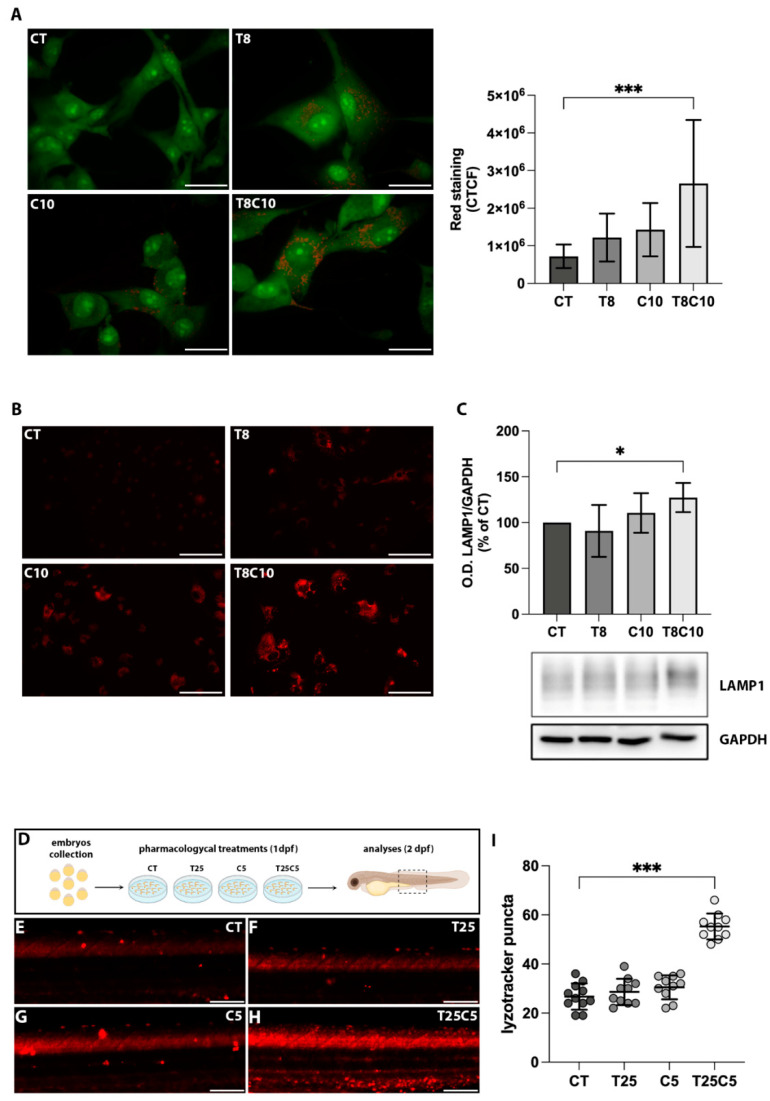
Combined HDAC6 and *Hh* inhibition altered acidic compartments. (**A**) U87-MG cells treated with vehicle (CT), 8 μM of TubA (T8), 10 μM of cyclo (C10), and 8 μM of TubA +10 μM cyclo (T8C10) for 48 h and stained with acridine orange. The AO accumulation into acidic compartments is represented by red dots. Green fluorescence arose from AO binding to DNA and RNA (**B**). Quantitative analysis of red fluorescence was calculated as corrected total cell fluorescence (CTCF) ± standard deviation. (**C**) LysoTracker Red staining in U87-MG cells treated with or without T8, C10, or T8C10. (**D**) Western blot analysis of LAMP1 protein expression normalized against GAPDH. Data are expressed as the percentage of the CT. (**E**) Schematic description of the pharmacological treatments in zebrafish. (**F**–**I**) Representative images of the trunk region of zebrafish embryos treated with 25 μM of TubA (T25) or 5 μM of cyclo (C5), alone or in combination, and stained with LysoTracker Red. (**I**) Quantification of the LysoTracker Red intensity in the trunk region of treated embryos. Scale bar indicates 40 μm (**A**) and 100 μm (**B**,**E**–**H**). Data are presented as mean ± standard deviation. CT—vehicle; TubA/T—tubastatin A; cyclo/C—cyclopamine. One-way ANOVA with Tukey post hoc correction. *** *p* < 0.001; * *p* < 0.05; nonsignificant data are not shown.

**Figure 4 ijms-24-05771-f004:**
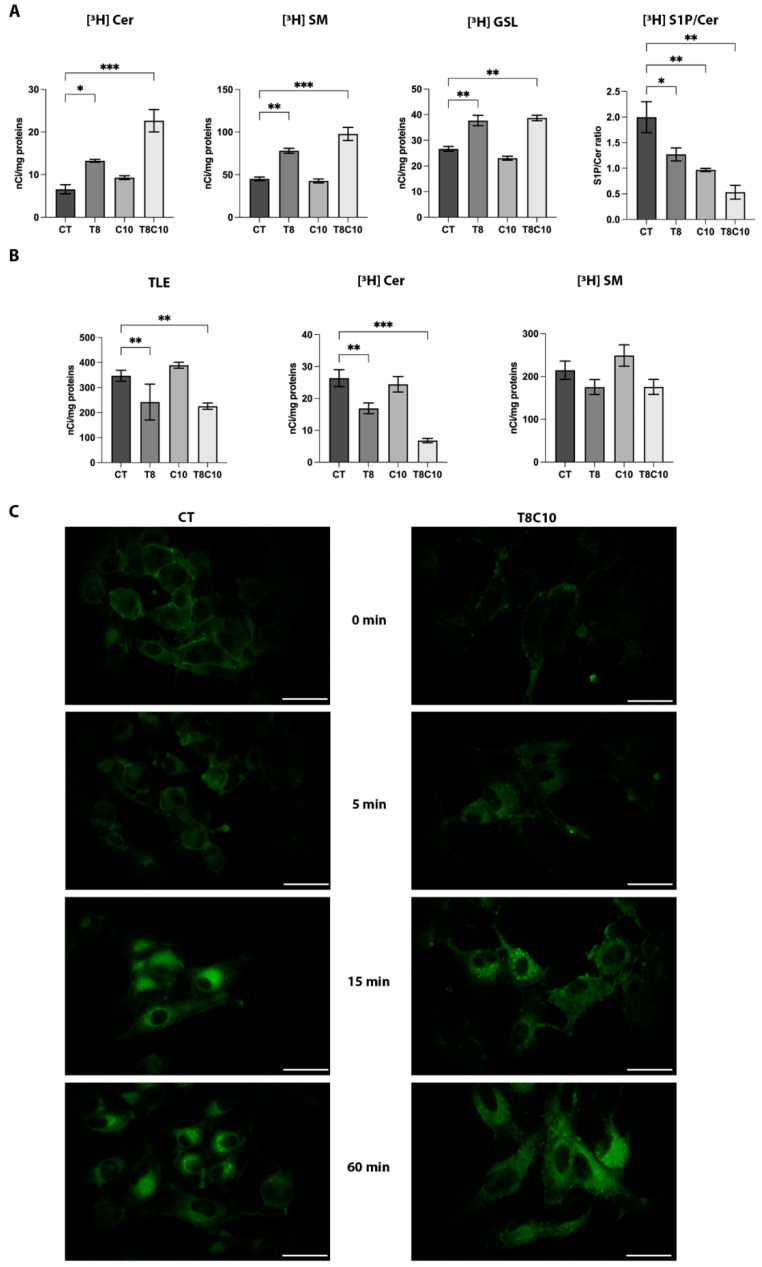
Sphingolipid metabolism and distribution were impaired following *Hh* and HDAC6 inhibition. (**A**) Radioactivity distribution among different sphingolipids after [1-^3^H]sphingosine administration at the metabolic steady state in U87-MG cells untreated (controls, CT) or treated with 8 μM of TubA (T8), 10 μM of cyclo (C10), and 8 μM of TubA + 10 μM cyclo (T8C10). Data are presented as mean of nCi/mg protein ± standard deviation. (**B**) Radioactivity of the total lipid extract (TLE), ceramide (Cer), and sphingomyelin (SM) following 2 h of [^3^H]SM administration in U87-MG cells CT or treated with T8, C10, or T8C10. Data are presented as mean of nCi/mg protein ± standard deviation. (**C**) Uptake and subcellular localization of fluorescent BODIPY-SM after treatment in U87-MG cells treated with or without T8C10 at 0 min, 5 min, 15 min, and 60 min. CT—vehicle; TubA/T—tubastatin A; cyclo/C—cyclopamine. One-way ANOVA with Tukey post hoc correction. *** *p* < 0.001; ** *p* < 0.01; * *p* < 0.05; nonsignificant data are not shown. Scale bar indicates 40 μm.

**Figure 5 ijms-24-05771-f005:**
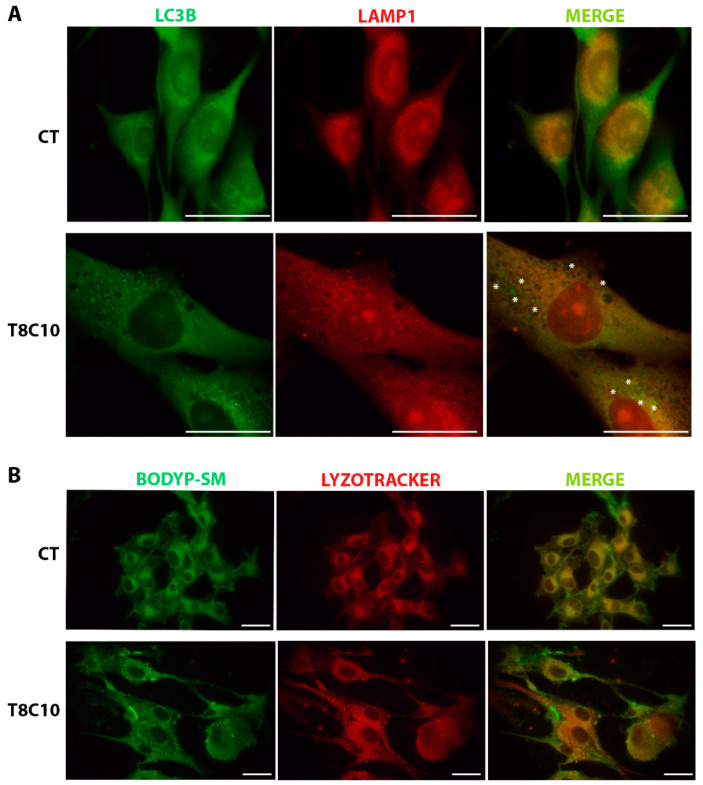
LC3 and LAMP1 colocalization and BODIPY-SM distribution were impaired following *Hh* and HDAC6 inhibition. (**A**) Double immunostaining against LC3 and LAMP1 of U87-MG cells treated in the presence or absence of both 8 μM of TubA and 10 μM of cyclo. Asterisks show some LC3 puncta, indicating autophagosomes, that did not colocalize with the red ones, indicating lysosomes. (**B**) BODIPY-SM and LysoTracker Red DND-99 staining of U87-MG cells treated in the presence or absence of both 8 μM of TubA and 10 μM of cyclo. CT—control, TubA/T—tubastatin A; cyclo/C—cyclopamine. Scale bar indicates 40 μm.

**Figure 6 ijms-24-05771-f006:**
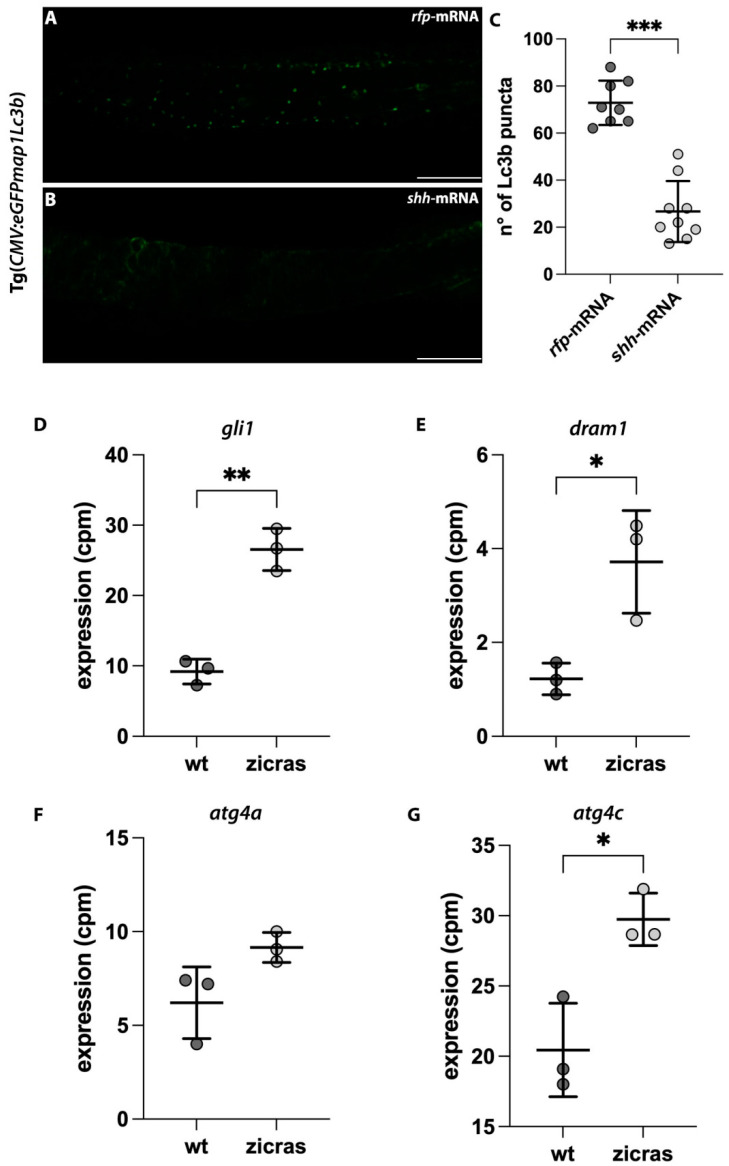
Impairment of the autophagic process in the zebrafish model with *Hh* hyper activation and in the zicras mutants. (**A**,**B**) Confocal images of the trunk-tail region of Tg(*CMV:eGFPmap1Lc3b*) embryos injected with (**A**) *rfp*-mRNA or (**B**) *shh* mRNA. (**C**) Count of the Lc3b puncta of *rfp-* and *shh*-mRNA injected embryos. Scale bar indicates 100 μm. (**D**–**G**) Expression analyses of (**D**) *gli1* and genes involved in autophagy activation, including (**E**) *dram1*; (**F**) *atg4a,* and (**G**) *atg4c,* derived from the analyses of wild-type and zicras (tumor) zebrafish brains (GSE74754). Data are presented as mean ± standard deviation. cpm—counts per million. Unpaired t-test with Welch correction. *** *p* < 0.001; ** *p* < 0.01; * *p* < 0.05. Scale bar indicates 100 μm.

**Figure 7 ijms-24-05771-f007:**
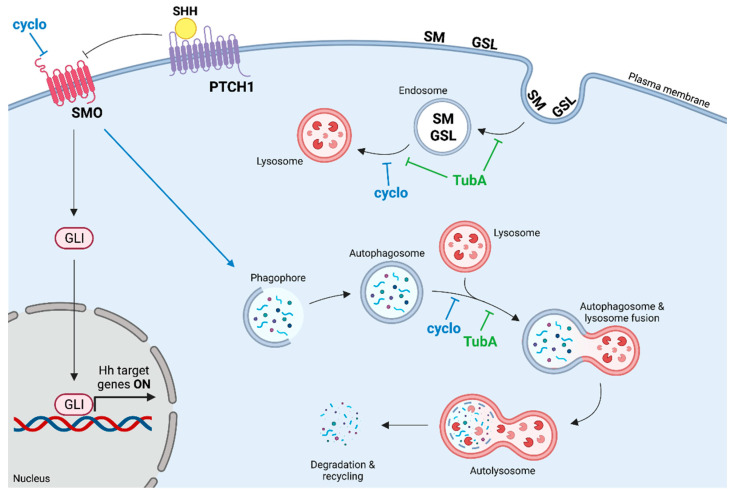
Mechanism of action of TubA and cyclo. The path of cyclo is indicated in blue while that of TubA is indicated in green. cyclo: cyclopamine; TubA: tubastatin A. SMO: smoothened; PTCH1: Ptached1; GLI: glioma-associated oncogene. SM: sphingomyelin, GSL: glycosphingolipid. Image generated with the BioRender.com tool.

## Data Availability

Data will be available upon request.

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
