# Peer review of "Combined Inhibition of Hedgehog and HDAC6: In Vitro and In Vivo Studies Reveal a New Role for Lysosomal Stress in Reducing Glioblastoma Cell Viability"

_ijms, 2023, doi:10.3390/ijms24065771_

Round 1
Reviewer 1 Report
Combined inhibition of Hedgehog and HDAC6: in vitro and in vivo studies reveal a new role for lysosomal stress in reducing glioblastoma cell viability
The study investigates the effects of combined inhibition of HDAC6 and the Hh signaling pathway, through TubastatinA and Cyclopamine respectively, in U87-MG human cells and zebrafish embryo. The approach and the overall design of the study are good. However, the authors should address the following concerns:
1. 2.1: “Data from the literature reported the overexpression of HDAC6 and the hyperactivation of the Hh signaling in GBM”. Reference missing.
2. Section 4.4: The rationale behind the selection of the dose for different pharmacological treatment should be mentioned
3. 4.6: Reference is missing for the labeling protocol of U87-MG cells
4. 4.8: Reference formatting needs checking
5. References are not properly cites for most of the methods section. Even if the protocols are conceived from previous studies with modifications, references should be cited and explain the modifications performed in the present study.
6. Please have the manuscript proofread by all contributors. There are grammatical and typographic errors at multiple instances.
7. Include future implications and limitations of the study.
Author Response
We thank the Reviewer for the revision and we provide the reply in the attached file.

Reviewer 2 Report
Well performed study
Author Response

(The authors gave the same response as above.)

Reviewer 3 Report
Review for: “Combined inhibition of Hedgehog and HDAC6: in vitro and in vivo studies reveal a new role for lysosomal stress in reducing glioblastoma cell viability”.
In the present study, the authors show that hedgehog and HDAC6 inhibition in glioblastoma cell line decrease lysosomal stress and reduced viability. Furthermore, the authors performed similar studies in zebrafish embryos and proposed a lysosome-dependent mechanism that involves autophagy and sphingolipid homeostasis. The presented manuscript is accompanied with thorough amount of data that helps support the authors ongoing hypothesis and definitely interesting, impactful and within the scope of this journal. However, some pending issues were identified within the presenting data and additional data should be presented prior to acceptance. Please read the following comments and please address the accordingly.
--Please include what the abbreviation stand for in the main body text the first time it is used (even if it has already been mentioned in abstract).
-“Hh signaling that enables GBM stem cells survival inducing chemo- and radiotherapy resistance”. There is no GBM stem cells, are the authors probably referring to neural stem cells or other type of stem cell? Furthermore, how does cell survival induce chemo and radio resistance? Please justify or emend this sentence.
- “CM-Dil and the transparency of the zebrafish embryos”. Zebrafish embryos are not transparent but translucent.
On Figure 1.
-Were the same number of cells injected in zebrafish embryos? How can the authors explain the fact that Figure D has substantially more cells/area than figure B but when they quantified, they were the same or lower? Authors should also provide figures of t0 injected embryos.
-Figure G. Housekeeping protein should be included in the western blot and data normalized to housekeeping proteins rather than erk and akt expression. This comment applies to all the other experiments in which housekeeping protein was not used as reference.
Figure 2.
-Do the authors have data or provide information that explains whether TubA and Cyclo can bypass the zebrafish chorion? This is crucial as most molecules cannot pass the choriorion hence the need to remove chorion in similar experiments.
-Figures E to M lack resolution and signal is difficult to observe. Can authors provide a better figure resolution in the document? Alternatively, please send figures in attachment for better visualization.
Figure 3.
-Image A and B lack scale bar
Figure 5.
-Images lack scale bar.
-In figure 3 authors reported that LAMP1 expression was significantly higher in T8C10 but on figure 5 A no apparent differences is observed on staining’s. Can the authors provide an explanation?
- The authors suggest a colocalization of the two markers, however it is premature to discuss that both proteins co-localize together as images lack resolution and the stains are expressed everywhere. Higher resolution or different approaches (i.e immunoprecipitation) need to be performed to infer such observations.
Figure 6.
-It is not clear what type of analysis was performed on images D to G
-As the authors inject rfp-mRNA can they provide images show RFP signal to show the success of microinjection?
Methods
-4.6 ref missing
-Were images generated through biorender? If yes credit should be given.
Author Response

(The authors gave the same response as above.)

Reviewer 4 Report
The manuscript, entitled "Combined inhibition of Hedgehog and HDAC6: in vitro and in vivo studies reveal a new role for lysosomal stress in reducing glioblastoma cell viability " This work is merited for publication in International Journal of Molecular after some major modification. So, I have some points that may help to improve the work as follows:
1-Abstract is good but need more explain about the main aim of work
2- The introduction should be extended to discuss the hypothesis and research questions in details. Additionally, the introduction should cover the recent literature related to this subject.
3- Material and methods
The methodologies should be explained in details so that the results are reproducible.
4-Results
The results are clear and important.
5-Discussion
The discussion section still needs improvement, and should be linked to the findings of the previous reports on this topic.
6- The conclusion
A section for conclusions need more explain and should include the most significant findings and future works only.
7- English writing should be checked by a native English-speaking expert.
Author Response

(The authors gave the same response as above.)

Round 2
Reviewer 4 Report
Authors have suitably revised the manuscript by addressing the reviewer comments and suggestions. This can be accepted for publication.